# EF-TTOA: Development of a UAV Path Planner and Obstacle Avoidance Control Framework for Static and Moving Obstacles

**Hongbao Du, Zhengjie Wang * and Xiaoning Zhang**

School of Mechatronical Engineering, Beijing Institute of Technology, Beijing 100081, China;
derek_dhb@bit.edu.cn (H.D.); xnzhang@bit.edu.cn (X.Z.)
* Correspondence: wangzhengjie@bit.edu.cn

**Abstract:** With the increasing applications of unmanned aerial vehicles (UAVs) in surveying, mapping, rescue, etc., the security of autonomous flight in complex environments becomes a crucial issue. Deploying autonomous UAVs in complex environments typically requires them to have accurate dynamic obstacle perception, such as the detection of birds and other flying vehicles at high altitudes, as well as humans and ground vehicles at low altitudes or indoors. This work's primary goal is to cope with both static and moving obstacles in the environment by developing a new framework for UAV planning and control. Firstly, the point clouds acquired from the depth camera are divided into dynamic and static points, and then the velocity of the point cloud clusters is estimated. The static point cloud is used as the input for the local mapping. Path finding is simplified by identifying key points among static points. Secondly, the design of a trajectory tracking and obstacle avoidance controller based on the control barrier function guarantees security for moving and static obstacles. The path-finding module can stably search for the shortest path, and the controller can deal with moving obstacles with high-frequency. Therefore, the UAV can deal with both long-term planning and immediate emergencies. The framework proposed in this work enables a UAV to operate in a wider field, with better security and real-time performance.

**Keywords:** obstacle avoidance; environmental features; unmanned aerial vehicle; control barrier function

## 1. Introduction

Unmanned aerial vehicles (UAVs) have been used more often in several industries in recent years, including surveying and mapping, rescue, patrol, reconnaissance and cluster attacks [1–5]. With the miniaturization of onboard computers and stereo cameras, UAVs can replace humans to complete complex tasks in harsh environments. Moreover, the rapid development of simultaneous localization and mapping (SLAM) has realized UAV autonomous flight in unknown environments [6,7]. As a result, the safe flight of UAVs has become a hot topic, and motion planning and control are inextricably linked to safety.

Recently, UAV motion planning has made great progress. The Fast planner [8–10] adopts a kinodynamic and topological method to search for a safe and feasible trajectory in the discretized control space of a quadrotor, achieving good experimental results in static environments. The topology-guided kinodynamic (TGK) planner [11] is a lightweight planner, but moving obstacles are not considered. The Ego planner [12] takes an optimistic approach to simplify the front-end path search, optimizing rough initial paths into feasible trajectories, making the algorithm more lightweight without using a Euclidean signed distance function (ESDF) map. In general, two main path planning methods include searching-based and sampling-based. Searching-based methods can often find the optimal path, but it takes more time in 3D grid maps [13–15]. The sampling-based method can quickly find the reachable path, but it is often a suboptimal solution [16–18].

Furthermore, some local planners concerning a dynamic environment have been proposed recently. The vision-based dynamic environment motion planning of quadrotors

by online replanning was realized in [19], but the framework for avoidance was limited by the frequency of the planner. The EB-RRT local planner, proposed in [20], performed real-time optimal motion planning for a mobile robot in a dynamic environment by combining the elastic band and RRT. The work improved the planner but presented more expensive computational complexities. As shown in Figure 1, a lightweight and reactive method is more applicable for safety when there is a moving obstacle blocking the origin trajectory. The UAV should have the ability to avoid obstacles whilst tracking the origin trajectory.

As for obstacle avoidance control of an unmanned system, there are many methods for safety verification. Two popular approaches are the Hamilton–Jacobi (HJ) reachable set and the control barrier function (CBF) [21,22]. As the dimensions of state space increase, computing the HJ reachability set becomes increasingly expensive, while the CBF offers real-time system safety with low computational complexity [22]. Therefore, the CBF has been applied in many fields, such as bipedal robots [23] and mobile vehicles [24]. However, the CBF cannot be directly used in high-order systems for its special property. For this reason, Q. Nguyen used the CBF for the safety of high-relative-degree systems, using back-stepping to derive the control inputs, which is complex and difficult [25].

As a result, we aim to establish a more generic architecture of planning and control to handle the common environment. On the one hand, incremental establishment of feature points of the static environment can make the path shorter and search faster. A more simplified front-end can be developed by detecting the environmental feature points to quickly find the shortest path. Concretely, corner points in a static environment are obtained by 3D point cloud feature extraction. In case the target cannot be reached directly, the shortest path always passes through the obstacle corners. Based on the construction of the boundary point graph, only the corner points become expanded when expanding the neighbour nodes. A small amount of data is maintained in an open list, saving memory for queue sorting and neighbour node expanding. On the other hand, the essential difference between planning and control is that motion planning has no feedback over a long period of time, which is an open-loop control, while control has feedback over a shorter period, which is closed-loop planning. Moving obstacles have uncertain displacement in a planning period, so the open-loop method is unsuitable. Figure 1 shows that moving obstacles in the local map occupy the original trajectory in the replanning horizon, and the UAV has no time to plan a new trajectory. The two main modules in this work are the environment feature-based (EF) planner and the trajectory tracking and obstacle avoidance (TTOA) controller.

The main contributions of this work are listed as follows:

(1)   EF planner

We proposed an efficient planner where front-end path finding is based on the environment feature points of obstacles, and the back-end trajectory optimization uses the convex hull property of a B-spline curve to ensure the safety of the trajectory from the corners of obstacles. Our planner can quickly find the shortest path and achieve a smooth and safe trajectory.

(2)   TTOA controller

Our developed controller is based on the CBF and can track the trajectory while avoiding moving obstacles in the environment with a small amount of computation without losing the original tracking.

(3)   EF-TTOA framework for a common environment

The EF-TTOA is a more general architecture that deals with the common environment. The new framework can make an autonomous UAV have both long-term planning ability and temporary adaptability.

The rest of the paper is arranged as follows. Section 2 introduces the detailed method of UAV planning and control in a common environment where static and dynamic obstacles exist simultaneously. Section 3 describes the implementation of the simulation experiment based on the robot operating system (ROS). Section 4 presents the conclusions from this work and briefly describes future work.

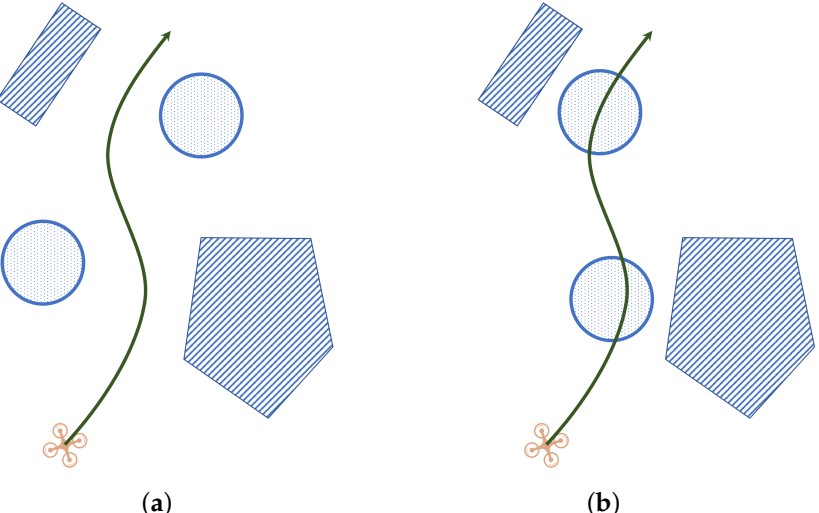

(**a**)　　　　　　　　　　　　　(**b**)

**Figure 1.** The original trajectory will be blocked by obstacles in the next moment. Circles filled with dots represent dynamic obstacles, while polygons filled with lines represent the static environment. (**a**) A smooth trajectory is given by the motion planner through the clear space safely. (**b**) Two dynamic obstacles blocking the original trajectory.

## 2. Methods

The goal of this work is to address a UAV's autonomous navigation in typical surroundings. We proposed a framework for planning and controlling a quadrotor UAV to achieve both target arrival and obstacle avoidance. The schematic diagram of the planning and control method flow is presented in Figure 2. Firstly, it is necessary to classify the detected continuous frames of point clouds into two classes, dynamic and static. The static point clouds are built as a map, while the dynamic point clouds are removed and used as the input for the trajectory tracking and obstacle avoidance (TTOA) controller after position and velocity estimation. Second, both the front- and back-end of the suggested EF planner have seen significant development. Path finding only focuses on a few key points that are boundary and corner points of the point cloud map. The back-end generates a B-spline curve based on the path and optimizes the trajectory by using the convex hull property. Finally, the TTOA controller is designed including trajectory tracking control and obstacle avoidance control based on the CBF. The TTOA controller can track the planned trajectory and avoid moving obstacles.

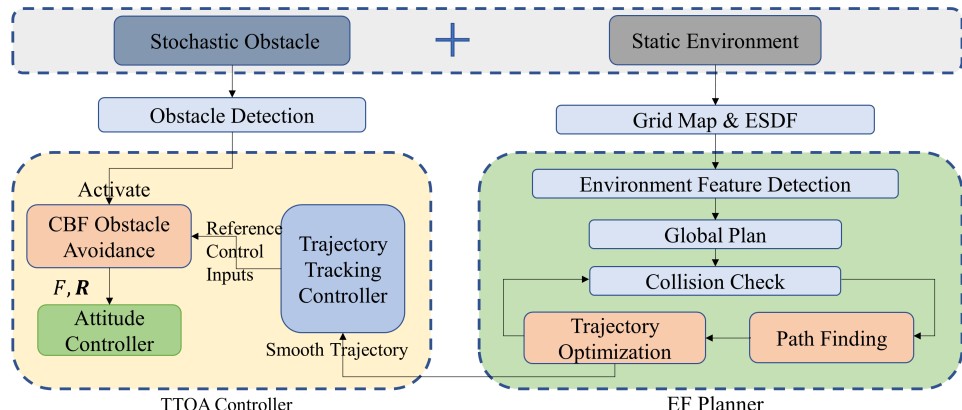

**Figure 2.** The proposed framework of the path planner and obstacle avoidance control for static and moving obstacles.

*2.1. EF Planner*

A requirement for the motion planning and control of the UAV is the perception of obstacles. Obstacles in the real world can be both moving and static. We use DBSCAN [26] and the voting of an individual point to classify obstacles based on several continuous frames of the point cloud [27]. The overall process is shown in Figure 3. First, the average velocity of each point cloud is calculated after ground filtering and down-sampling. Then the velocity of each point cloud cluster is obtained by Equation (1), where $v_i$ is the velocity of obstacle, $t$ is time, $p_t$ and $p_{t-1}$ represent the position in current and previous frame, respectively, $n$ is the number of points contained in the cluster, and $\Delta t$ represents the time interval between two frames. When the point cloud velocity $v_i$ is greater than $v_{max}$ (0.3 m· s$^{-1}$), the object is categorized as a dynamic object because the UAV is already at risk from it. Furthermore, if a point's velocity exceeds $v_{max}$, the point will vote for it to be dynamic. The obstacle will be recognized as a dynamic obstacle if the ratio of dynamic votes $N_{vote}$ over valid points $N_{valid}$ is higher than another threshold $D_{ratio}$, represented in (2). Finally, two classifications of dynamic and static obstacles are obtained.

$$v_i = \frac{1}{n} \sum \frac{p_t - p_{t-1}}{\Delta t} \tag{1}$$

$$D_{ratio} = \frac{N_{vote}}{N_{valid}} \tag{2}$$

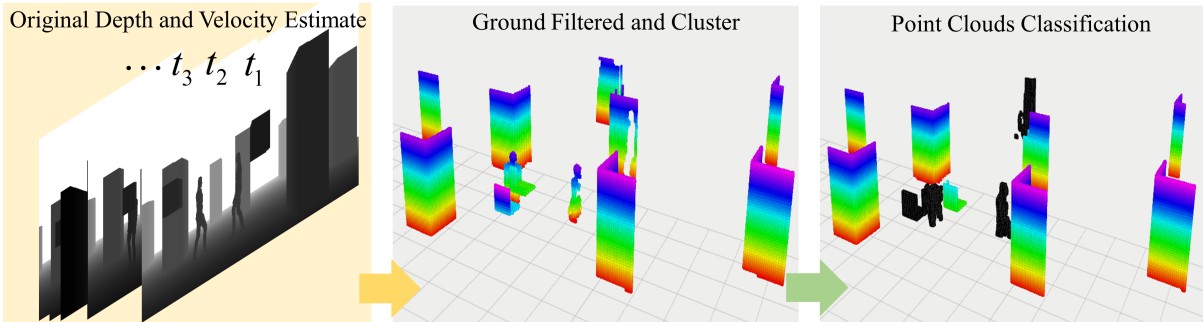

**Figure 3.** The process of point cloud clustering and classification.

2.1.1. Environment Feature Detection

The planner processes the static point cloud following the aforementioned classifications, while the controller handles the dynamic point cloud due to its temporal variability. The obstacle space is much smaller than the free space, so our work just focuses on the key points of static environments in path searching. Inspired by time elastic bands (TEB) [28,29] and jump point search (JPS) [30,31], a new front-end local planning approach is proposed. Similar to the elastic band, Figure 3 shows that the path stretches into a straight line when there are no obstacles between the starting point and the target. The elastic band will always go around some obstacle corners when the start point and target are blocked. Therefore, the planner includes two parts, the extraction of environmental feature points and the design of a path planner based on these points. To efficiently identify the shortest path in a complex environment, the system must concentrate on the boundary feature points of obstacles. By constructing a graph based on these points, we can perform a graph search algorithm to find the optimal path Figures 4 and 5.

Feature extraction for the down-sampled occupancy grid map can greatly reduce the search time. Only the edge and corner information is taken into account when determining the actual path, while the point cloud occupation is used as the gradient information for the three-dimensional convolution with the Prewitt kernel [32]. By selecting the appropriate kernel, the key feature points of the environment can be effectively obtained. Algorithm 1 demonstrates the entire proposed environment feature detection system for motion planning. To extract features from a grid map's point cloud, a sliding window is

used to convolve in three dimensions, moving in different directions. This process computes the point gradient changes within the window and ultimately calculates each point's Harris response value. These values allow for the characterization of each individual point in the point cloud. The occupation map is a binary map that distinguishes between obstructed and free configuration spaces. In this type of map, areas with obstacles are represented by "true" while "false" represents free space. Subsequently, the occupation gradient is calculated so that the gradient changes of each point are summed in Equation (3).

$$S(x,y) = \sum_u \sum_v w(u,v)[I(u,v) - I(u+x, v+y)]^2 \tag{3}$$

The gradient change can be simplified by the first two items of the Taylor expansion and then expressed as a difference by (4).

$$I(u+x, v+y) \approx I(u,v) + \frac{dI}{dx}(u,v)x + \frac{dI}{dy}(u,v)y \tag{4}$$

Therefore, the occupation gradient of this point changes as shown in Equation (5).

$$S(x,y) \approx \sum_u \sum_v w(u,v)[\frac{dI}{dx}(u,v)x + \frac{dI}{dy}(u,v)y]^2 \tag{5}$$

The covariance matrix $M$ is constructed after the gradient change in each direction of the obstacle point cloud is obtained. The response value $R$ of points can be obtained through matrix $M$, where $R = det(M) - k \cdot trace(M)^2$, and sparse corner points can be obtained by further filtering, as shown in Algorithm 1.

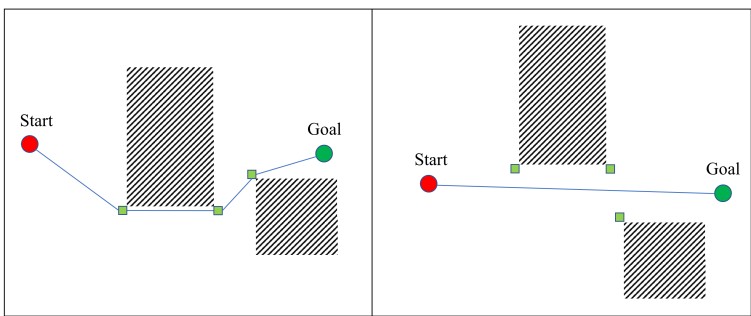

**Figure 4.** The shortest path between two points.

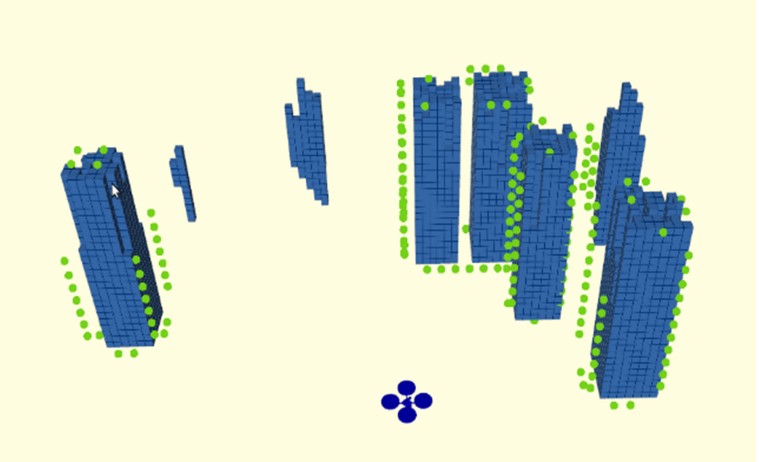

**Figure 5.** Feature extraction of environmental point cloud information can quickly provide accurate information for path searching. Blue blocks represent obstacles occupying the grid map and green points represent feature points of obstacles.

---

**Algorithm 1** Environment feature detection

---

    **Result:** Write here the result
    Pos ← ConnerDetection(Start);
    Target←Initial();
    PCs = ∅;
    FeaturePoints=∅;
    **while** *Pos != Target* **do**
        PCs←Subscribe(Local Point Clouds within FOV) ;
        PCs.DownSampling () ;
        Convolution3D(PCs);
        [Nx,Ny,Nz]←NormalVector(PCs);
        **for** *point* ∈ *PCs* **do**
            $M$←Construct(Nx,Ny,Nz);
            $R$←ResponseOfDetector($M$);
            Non-MaxThreshold($R$);
            FeaturePoints.push(point);
        **end**
    **end**

---

### 2.1.2. Path Planning

In contrast to the random search method of local sampling, this work establishes the sparse search space with points as a visual graph on the basis of accurately obtaining the potential waypoints and then uses the heuristic search algorithm to finally obtain the shortest path. This work uses Algorithm 2-based A* for path finding. P is the feature points set and G is the graph constructed by these points. Unlike the traditional A*, the neighbour node of each point is not the neighbour node of the grid map but all the visual nodes after collision detection at a certain point. In addition, the global search space can be stored for a return trip. It is unnecessary to repeatedly map and search the explored environment as it can be reused with one accurate exploration.

---

**Algorithm 2** Path finding

---

    P ← ConnerDetection(Start);
    G ← CreateGraph( P,Target);
    OPEN = ∅;
    CLOSE = ∅;
    OPEN.push(START) ;
    **while** *OPEN != ∅* **do**
        N ← OPEN.popMin() ;
        CLOSE.push(N);
        **if** *N=Target* **then**
            **return** true;
            **break** ;
        **else**
            **for** *M* ∈ *unexpanded neighbors of N* **do**
                **if** *g(M)=infinite* **then**
                    $g(M) \leftarrow g(N) + C_{NM}$ ;
                    OPEN.push($N$);
                **else if** *g(M) > g(N) + $C_{NM}$* **then**
                    $g(M) \leftarrow g(N) + C_{NM}$ ;
                **end**
            **end**
        **end**
    **end**

---

2.1.3. Trajectory Generation and Optimization

Because this cannot be fed directly to the UAV for operation after using the above method to find the shortest path, it is necessary to generate a safe, smooth and dynamic feasible trajectory. The trajectory must be dynamically feasible for the UAV and collision-free. We use the uniform B-spline curve as the UAV trajectory. Because of its benefits, we utilize a B-spline curve for the UAV's trajectory. For instance, the convex hull property and features of a B-spline derivative are still a B-spline curve. Both of these properties effectively assist us to ensure that the trajectory is collision-free during gradient optimization. The former property makes it simpler for us to impose the dynamic constraints of the UAV on the trajectory. The three basic elements of a B-spline curve are the degree $P_b$, control points $\{Q_0, Q_1, \ldots, Q_N\}$ and knot vector $[t_0, t_1, \ldots, t_M]$. The complete description of the B-spline curve used here is as defined in Equation (6).

$$
\begin{aligned}
p(s(t)) &= s(t)^T M_{p_b+1} q_m \\
s(t) &= [1 \quad s(t) \quad \ldots \quad s^{pb_t}(t)]^T \\
q_m &= [Q_{m-P_b} \quad Q_{m-P_b+1} \quad Q_{m-P_b+2} \quad \ldots \quad Q_m]
\end{aligned}
\tag{6}
$$

where $p_b$ set as 3 and $M_{p_b}$ is a constant matrix [33], $Q_N \in R^3$. The cost function is defined in Equation (7).

$$
J = J_c + J_d
\tag{7}
$$

where $J_c$ and $J_d$ ensure the safety and dynamic feasibility of the trajectory, respectively.

The trajectory optimization is based on the ESDF map. In order to ensure the smoothness and safety of the trajectory, we transform the trajectory optimization of this work into an optimization problem with inequality constraints, as shown in Equation (8). The smoothness of the trajectory is the goal of the objective function. Using the B-spline's convex hull property, the constraint inequality makes sure that the trajectory close to the obstacle's corner does not encroach into its interior.

The convex quadrilateral produced by four knots on either side of the corner point is outside the obstruction, according to the inequality constraint, guaranteeing the safety of the associated stage. Figure 6 illustrates the steps taken to develop the initial trajectory into a feasible trajectory. The green line shows the optimum safety trajectory, while the red arrows indicate the gradient field formed by obstacles that push the orange initial trajectory outward.

$$
J_c = \sum_{i=p_b-1}^{N-p_b+1} \| Q_{i+1} - 2Q_i + Q_{i-1} \|
\tag{8}
$$

$$
s.t.(AQ_{i:i+3} - BQ_j) \cdot \nabla_{esdf}(Q_j) \geq 0
$$

where $Q_j$ is the corner point in the initial path and $\nabla_{esdf}(Q_j)$ represents the gradient vector of the ESDF map at the corner point. The matrices $A$ and $B$ represent the convex hull and are defined below.

$$
A = \begin{bmatrix} 1 & 0 & 0 & 0 \\ 0 & 0 & 0 & 1 \\ -1 & 1 & 0 & 0 \\ 0 & 0 & 1 & -1 \end{bmatrix}, B = \begin{bmatrix} 1 \\ 1 \\ 0 \\ 0 \end{bmatrix}.
$$

$J_d$ is an additional cost function ensuring the dynamic feasibility of the UAV, as shown in Equation (9). This achieves the dynamic feasibility of the entire trajectory by punishing points that exceed the UAV's velocity and acceleration control points.

$$
J_d = \sum_{i=p_b-1}^{N-p_b} f_d(V_i) + \sum_{i=p_b-2}^{N-p_b} f_d(A_i)
\tag{9}
$$

where $f_d(v) = max(0, v^2 - v_{max}^2)^2$, $V_i = \frac{Q_{i+1}-Q_i}{\Delta t}$ and $A_i = \frac{Q_{i+1}-Q_i}{\Delta t}$ represent the velocity and acceleration at each trajectory control point, respectively.

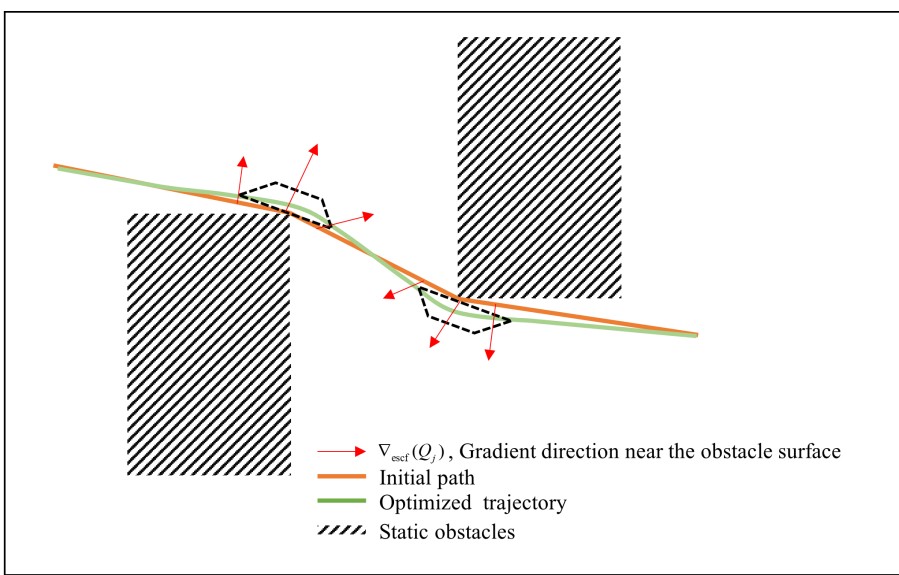

**Figure 6.** The brown line is the initial path, the green curve is the optimized path and the red arrows are the gradient direction formed by the obstacles. The black dotted lines are the convex hull ensuring the safety of the trajectory.

### 2.2. TTOA Controller

In this section, the TTOA controller is developed from the CBF based on quadrotor dynamics. First, the basic theory of the CBF is introduced and a more common method is developed based upon it. Second, the TTOA controller is designed for flexible trajectory tracking and obstacles avoidance.

#### 2.2.1. Control Barrier Function

Firstly, a non-linear affine control system is considered in Equation (10).

$$\dot{x} = f(x) + g(x)u \tag{10}$$

where $x \in \mathbf{R}^n$, $f : \mathbf{R}^n \to \mathbf{R}^n$ , $g : \mathbf{R}^{n \times m} \to \mathbf{R}^n$ are locally Lipschitz. $u(x,t)$ is the input of the system. Similar to the control-Lyapunov function (CLF) expanded from the Lyapunov function, the CBF is an extension-form barrier function to systems with control inputs. The ordinary Lyapunov function is used to verify a system's stability while CLF is used to find feasible control such that the system can be brought to the zero state asymptotically. The CBF is used to limit the controller so that the system state $x$ does not enter domain $D$. Therefore, the difference is that the former ensures the stability of the system, while the latter ensures the safety of the system. Considering affine systems (10), the main role of the CBF is to construct a safe set described as the barrier function $h(x) : D \subset \mathbf{R}^n \to \mathbf{R}$, yielding Equation (11):

$$\begin{aligned} C &= \{x \epsilon D \subset \mathbb{R}^n : h(x) \geq 0\} \\ \partial C &= \{x \epsilon D \subset \mathbb{R}^n : h(x) = 0\} \\ Int(C) &= \{x \epsilon D \subset \mathbb{R}^n : h(x) > 0\}. \end{aligned} \tag{11}$$

By designing a suitable control input invariant set, the system state is limited to the safe set $C$. Let $C \subset D \subset \mathbf{R}^n$ be the super-level set of a continuously differentiable function $h : D \to \mathbf{R}$, then $h$ is a CBF if an extended class $K_\infty$ function $\gamma$ exists for the control system (Figure 7).

$$\sup_{u \in U} \left[ L_f h(x) + L_g h(x)u \right] \leq \gamma(h(x)) \tag{12}$$

where $\dot{h}(x) = L_f h(x) + L_g h(x) u$ for all $x \in D$. Usually, a CBF-QP is used to form a quadratic programming (QP) problem as shown in Equation (13).

$$u(x) = \arg \min_{u \in R^m} \frac{1}{2}(u - u_n)^T H(u - u_n)$$
$$s.t. \left[ L_f h(x) + L_g h(x) u \right] \geq -\gamma(h(x)) \tag{13}$$

As a powerful tool for system safety, the CBF requires that the first derivative of the barrier function must explicitly include control input $u(x, t)$, which cannot be directly applied to high-order systems. A high-relative-degree constraint means that the $n$-order derivative of the obstacle function contains a control input, $n \geq 2$. Therefore, this work proposes a new type of CBF for quadrotors by using the relative velocity and position between the UAV and obstacles.

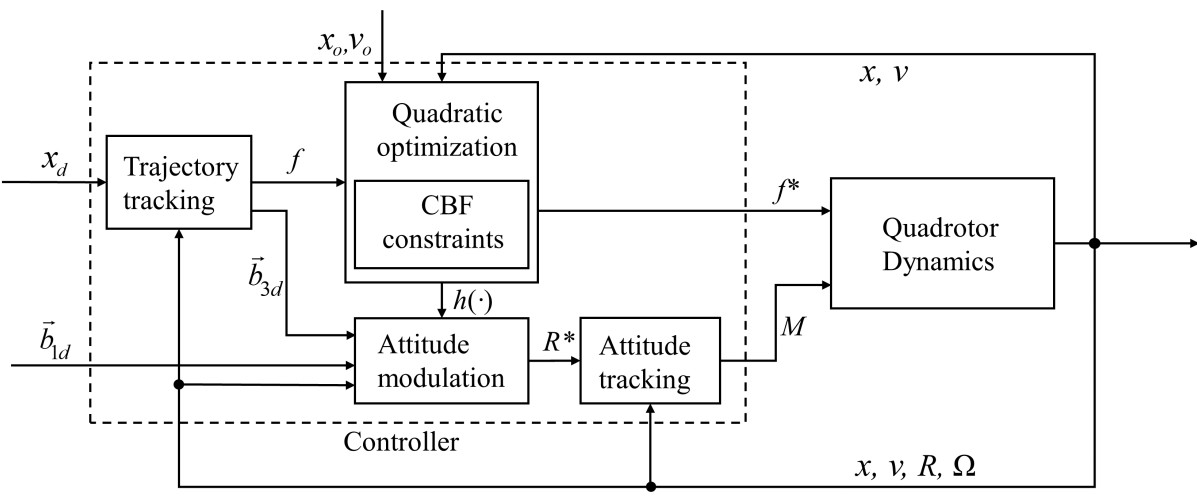

**Figure 7.** Block diagram of the closed-loop system.

### 2.2.2. TTOA Controller Design

The translation and rotation of a quadrotor, described as the Newton–Euler function, is shown in Equation (14). The UAV's position is controlled by changing its thrust $f \in \mathbf{R}$ and altitude $R \in SO(3)$, controlled by moment $M \in \mathbf{R}^3$.

$$x = \dot{v}s.$$
$$m\dot{v}s. = mg\mathbf{e}_3 + fR\mathbf{e}_3$$
$$\dot{R} = R\hat{\Omega} \tag{14}$$
$$J\dot{\Omega} + \Omega \times J\Omega = M$$

where $x \in \mathbf{R}^3$ and $v \in \mathbf{R}^3$ are the position and velocity of the UAV, respectively, and $f$ is the thrust of the UAV. $\Omega \in \mathbf{R}^3$ represents the angular velocity of the UAV, $J \in \mathbf{R}^{3 \times 3}$ is the inertia matrix with respect to the body-fixed frame. $e_3 = [0; 0; 1] \in \mathbf{R}^3$ is z-axis of the inertial frame. The hat map $\hat{\cdot} : \mathbf{R}^3 \rightarrow so(3)$ aims to convert a vector into an anti-symmetric matrix.

In contrast to the traditional tracking controller [34], the controller in this work needs to deal with static and moving obstacles on the track. The block diagram of the closed-loop system is shown in Figure 7. The main work this section describes the controller's design. $x_d$ and $\mathbf{b}_{1d}$ are the desired position and yaw obtained from the trajectory, respectively. Quadratic optimization is used to constrain the thrust of the UAV to ensure safety when facing obstacles. Before this, the desired roll angle of the UAV is altered according to the barrier function $h(\cdot)$. The new desired altitude $R^*$ is used as the reference input of the altitude tracking controller. Finally, trajectory tracking, quadratic optimization

constrained by the CBF, altitude modulation and altitude tracking constitute the complete TTOA controller.

In order to avoid moving obstacles, the CBF needs to contain motion information of the obstacle. Therefore, a velocity obstacle method is introduced into the CBF to constrain the velocity of the UAV. The CBF proposed this work can be defined by Equation (15)

$$h(x) = v_r^T H v_r - r_o^2 v_r^T v_r \tag{15}$$

where $H = [d^T d I - d d^T]$ is a symmetric matrix, constructed from the distance between the UAV and the obstacle. $r_o$ is the radius of the obstacle contour. $v_r$ is the velocity of the UAV relative to the obstacle. $h(x) \geq 0$ indicates that $v_r$ is falling into the obstacle area during flight. Through the Lie derivative of $h(x)$, we can obtain $\dot{h}(x)$ from Equation (16).

$$\dot{h}(x) = 2 H v_r \dot{v}_r + v_r^T \dot{H} v_r - 2 r_o^2 v_r^T \dot{v}_r \tag{16}$$

where $v_r = g e_3 - \frac{f}{m} R^* e_3$ , $\dot{H} = v_r^T d + d^T v_r - 2 d v_r^T$. $R^* = R_d R_\theta$ presents rolling based on the original expected altitude, while $R_d$ and $R_\theta$ are the desired altitude and rotation matrices of the UAV, respectively. According to the dynamics of the trajectory, we can adjust the feasible roll of the UAV to avoid obstacles. $c$ and $s$ represent $cos(\vartheta)$ and $sin(\vartheta)$, respectively. $\vartheta$ is a function of $h(x)$ and $b$ is a sign function describing the left or right roll of the UAV, obtained according to the original trajectory. Roll satisfies the dynamic constraints of the UAV by adjusting $l$ and $k$.

$$R_\vartheta = \begin{bmatrix} 1 & 0 & 0 \\ 0 & c & -s \\ -1 & s & c \end{bmatrix}, \vartheta = \frac{b\pi}{2(1 + e^{l - kB(x)})}$$

Therefore, by rewriting Formula (13), trajectory tracking and obstacle avoidance can be transformed into a QP problem by Equation (17). To avoid obstacles in real time, the optimal control must be solved online at each step. It is easy to find a closed-form solution to the QP program. For a low-dimensional QP with clear physical meaning, finding a closed-form solution greatly improves the real-time performance. In this work, the objective function is the quadratic function concerning thrust. As shown in Figure 8, gradient zones $O_i$ represent constraints caused by obstacles. $f \in [0, f_{max}]$ is the feasible control space and the approach of each obstacle will form a constraint to the UAV. The vertical lines are far from the nominal thrust $f_n$, when no obstacles are encountered. The constraints of the optimization problem are not activated, so the optimal control is the nominal inputs. As the obstacle approaches, the feasible control input space decreases as the lines move closer, and the optimal solution occurs at the lowest intersection of the lines and the quadratic curve.

$$u(x) = \arg \min_{u \in R^m} \frac{1}{2} (f - f_n)^2$$
$$s.t. \left[ L_f h_i(x) + L_g h_i(x) u \right] \geq -\gamma(h_i(x)) \tag{17}$$
$$f \leq f_{max}$$

where $i$ indicates the serial number of the obstacles. Finally, the TTOA controller contains two parts: thrust and attitude. A reasonable expected altitude $R^*$ can expand the feasible range of the UAV's thrust. In general, this can improve the obstacle avoidance ability of the UAV.

$$f^* = f_n - \frac{max(L_g h_i(x) f_n - \gamma h_i(x))}{L_g h_i(x)} \tag{18}$$
$$R^* = R_d R_\vartheta$$

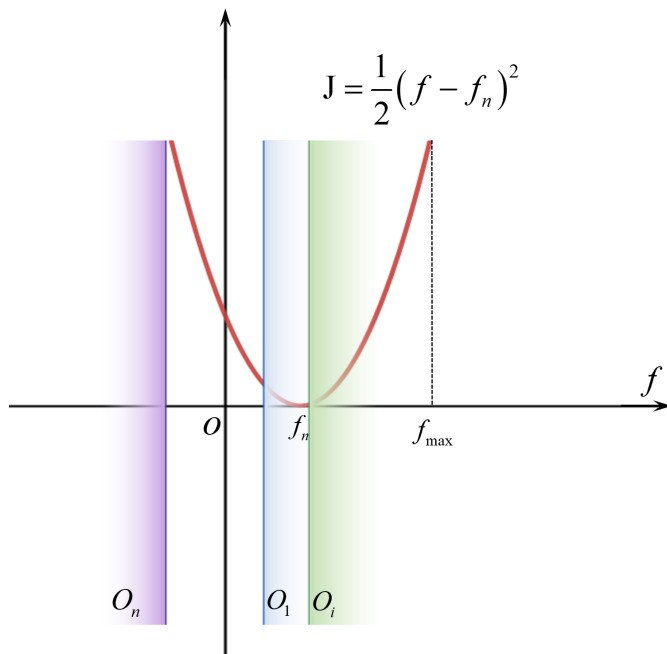

**Figure 8.** Illustration of QP constrained by the CBF.

## 3. Simulation and Discussion

We simulated the EF planner and TTOA controller with static and mixed environments, respectively. The static environment was randomly generated by a point cloud map kit for random tests. The more general situation included dynamic and static obstacles, simulated in the ROS-Gazebo-PX4 SITL environment. Figure 9 shows the simulation framework in the general obstacle environment. All modules were simulated on an Intel Core i7-10700F CPU@2.9GHz with 16 G SSD.

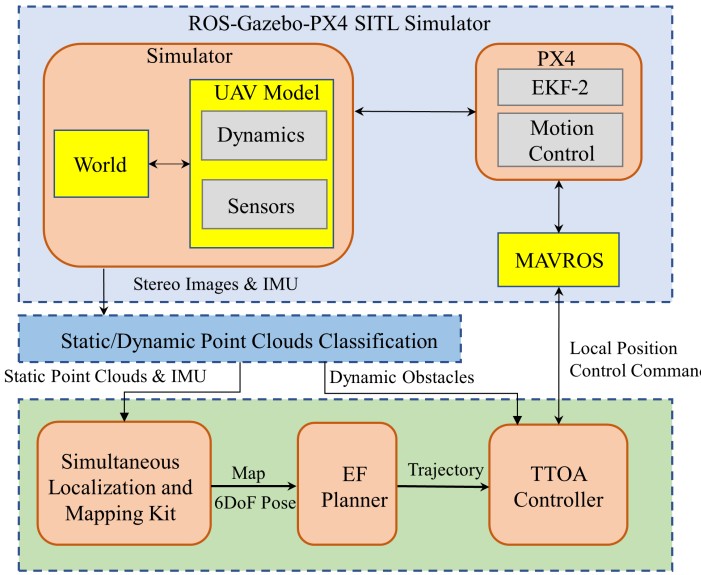

**Figure 9.** Framework of the simulation.

### 3.1. Simulation of the EF Planner with a Static Environment

The EF planner proposed in this work is implemented in C++11 with an open project 3D point cloud processing point cloud library (PCL). Feature extraction and path search is based on down-sampled environmental obstacle point clouds. The back-end trajectory

optimization adopts the B-spline optimization method and uses the gradient of the ESDF map to push the trajectory to a more spacious position.

In Figure 10, the blue blocks are obstacles in the environment, the green points are the corresponding feature points, and the purple disks and curve represent the UAV and its trajectory, respectively. The UAV marks and stores the edges and corners of the obstacle when continuously exploring the unknown environment, and the environmental feature points on the whole flight path are saved in a small storage memory.

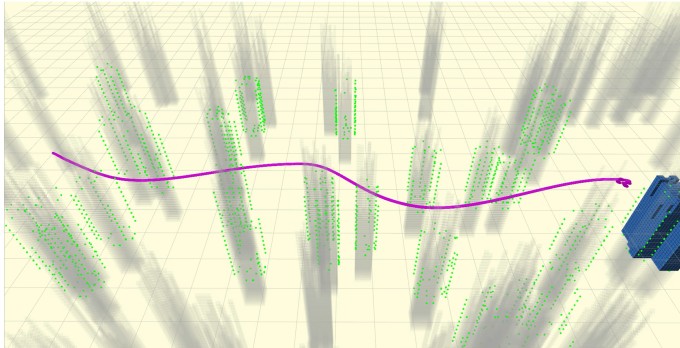

**Figure 10.** The quadrotor UAV autonomous flight with the proposed planner in dense environments.

The trajectory of the UAV bypassing obstacles is continuous and smooth. Figure 11 shows the comparison of time consumption for path finding and the length of the trajectory from (0,0,1) to (5,0,1) with different obstacle densities. Each group of data is the average of 10 tests. The path found by the proposed method in this work is obviously the shortest. The proposed planner takes less time than the state-of-the-art planners as the barriers become sparse. The EF planner can quickly search for the shortest path and generate a trajectory within 10 ms. Compared with the planner in [9], only feature points are considered in the process of path finding, reducing the sampling time and increasing the accuracy of the path. The search space of the proposed method is reduced to a sparse set of points, thus reducing the search time. However, as the density of obstacles increases, the proposed method requires more time to process the point cloud of features. Therefore, when the obstacle density is greater than 0.25, the proposed method takes more time than [9]. Therefore, the proposed method is unsuitable for situations where obstacles exceed 0.4. Generally speaking, 0.4 is already an extremely large number of obstacles.

### 3.2. Simulation of the Proposed Framework with a Mixed Scene

In order to verify the performance of the EF-TTOA framework in mixed scenarios, we realized this simulation with the ROS-Gazebo-PX4 open-source platform. As shown in Figure 12, the simulation environment is built in Gazebo. The UAV needs to cross the obstacle-filled environment from the initial position (0,0,1) to the goal (45,0,1.25). There is no effective way to deal with moving people and coloured cubes in the environment using the existing planners. The EF-TTOA framework can improve the success rate of the task.

Figure 13 shows the comparison of the actual flight trajectory with [9,12]. The planned and real trajectory of the proposed work are red and blue, respectively. The planner proposed in [9] fails to deal with moving dynamic obstacles, while the planner proposed in [12] always flies with a longer trajectory. It can be seen that the UAV can not only fly smoothly in the static environment, but also fly safely in the environment with moving obstacles. Figure 14 shows the position and yaw of the UAV. The simulation shows that our approach is able to find the shortest and smoothest path compared with the existing planners. In addition, our method is more applicable because the dynamic obstacle avoidance problem is delegated to the obstacle avoidance controller with higher-frequency. Compared with existing methods, our method has the advantage of a stronger ability to deal with a dynamic environment. In the simulation, it was found that adding roll angle modulation can more easily realize the obstacle avoidance-based CBF.

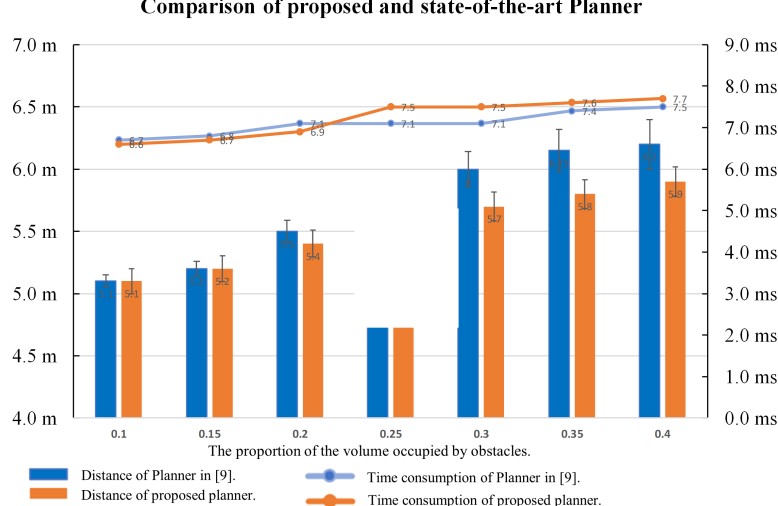

**Figure 11.** Comparison of the proposed and state-of-the-art planners in terms of time and distance, varying the obstacle density. The horizontal axis represents the proportion of the volume occupied by obstacles in the space.

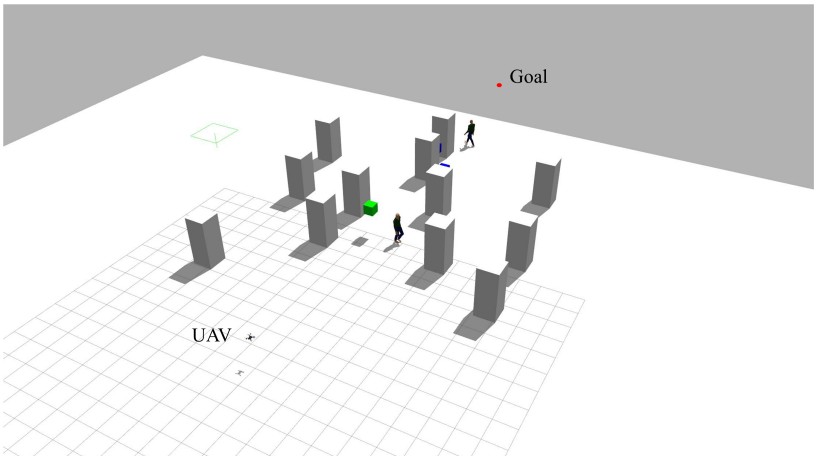

**Figure 12.** Task environment in the ROS-Gazebo-PX4 SITL simulator.

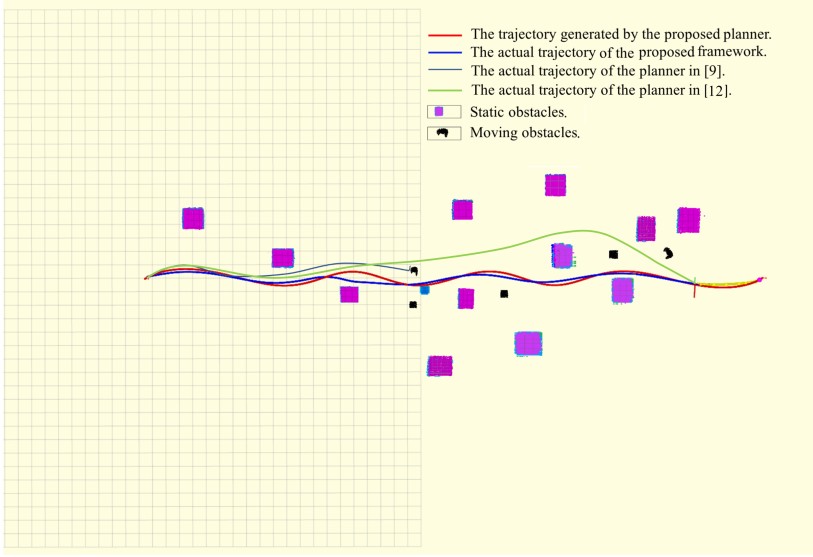

**Figure 13.** Comparison of the actual flight trajectory with [9,12].

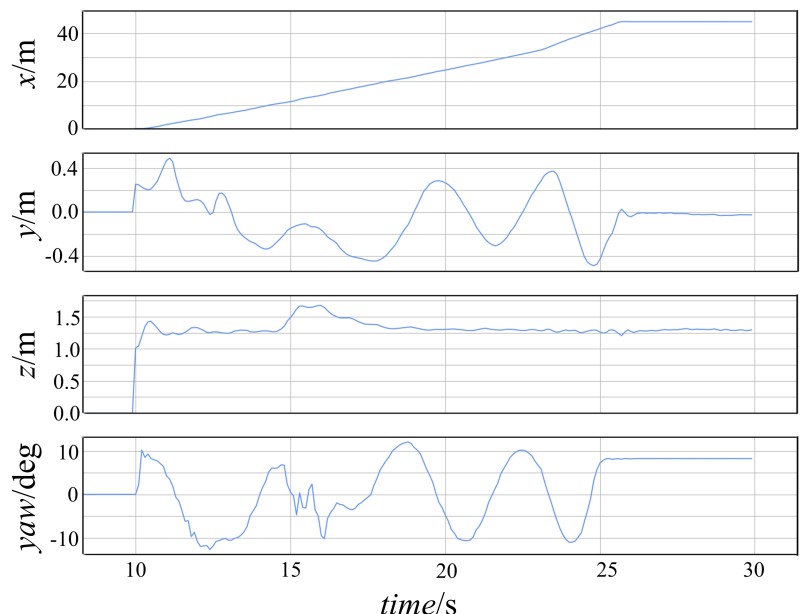

**Figure 14.** Position and yaw of the UAV.

## 4. Conclusions

In this work, a systematic planning and control technique for autonomous and flexible obstacle avoidance for UAVs is proposed. The EF planner constructs a fast front-end with a smaller search space and back-end with concise optimization for smoothness and corner collision-free trajectories. The control input is obtained by solving the CBF-QP problem with a closed-form solution. For moving obstacles, the controller rather than the planner performs the actions faster. Therefore, the EF-TTOA framework can handle more general scenarios. This technology is of great significance to other UAVs.

For further consideration, the method will be extended to fixed-wing UAVs for flying past obstacles, similar to a bird moving sideways or contracting its wings through a narrow channel.

**Author Contributions:** Conceptualization, H.D. and Z.W.; methodology, H.D.; project administration, H.D. and Z.W.; software, H.D.; supervision, X.Z. and Z.W.; validation, Z.W. All authors have read and agreed to the published version of the manuscript.

**Funding:** This research was funded by the Armament Pre Research Project Foundation of China, grant number 627010702.

**Institutional Review Board Statement:** Not applicable.

**Informed Consent Statement:** Not applicable.

**Data Availability Statement:** Not applicable.

**Conflicts of Interest:** The authors declare no conflict of interest.

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
