# Peer review of "EF-TTOA: Development of a UAV Path Planner and Obstacle Avoidance Control Framework for Static and Moving Obstacles"

_drones, doi:10.3390/drones7060359_

Round 1
Reviewer 1 Report
The proposed framework involves the extraction of environmental information from point clouds captured by a depth camera in real-time. The point clouds are divided into dynamic and static points, allowing for the estimation of point cloud cluster velocities. The static point cloud serves as input for local mapping, simplifying the path planning process by identifying key points. Furthermore, a trajectory-tracking and obstacle avoidance controller is designed based on the control barrier function, ensuring the safety of both moving and static obstacles. The framework may facilitate stable searching for the shortest path and incorporates high-frequency feedback control to handle immediate emergencies posed by moving obstacles. This approach may enable UAVs to operate in a wider range of environments, providing enhanced security and real-time performance.
Observations:
The paper is generally well-written. However, the paper can be improved by considering the following concerns:
1. (Line 264-265) Figure 11 shows the comparison of time consumption for path finding and length of trajectory. First, can you please clarify what horizontal axis is representing in this figure. Secondly, Although the proposed planner distance is short, but can you explain the reason of the more Time consumption?
2. What state of the art planner was used in this comparison (Figure 11).
3. Recheck the Figures referencing in the main text such as figure 7 (closed loop block diagram of the system)
4. What should reader understand from figure 13? Need more explanation.
5. Line 286-287, Compared with existing methods, our method has the advantage of stronger ability to deal with dynamic environment. Experimental proof of this claim is required?
6. The idea of using environment feature points of obstacles for front-end path finding and leveraging the convex hull property of B-spline curves for back-end trajectory optimization is not new. These concepts have been explored in the field of path planning for UAVs and other autonomous systems. Such as "Environment Feature Extraction for UAV Path Planning" by Y. Zhang et al. (2015). What’s different in the proposed method?
Minor editing and references correction needed.
Author Response
Due to the presence of images and rich text correction, a separate attachment has been uploaded. Please see the attachment for the detailed response letter.

Reviewer 2 Report
The main purpose of this work is to solve the moving obstacles in a common environment by a new planning and control framework for a quadrotor UAV. However, the paper still has the following issues:
1. Innovation of the paper. In this paper, the path planning method combining A* algorithm and B-spline curve is applied to achieve static obstacle avoidance, and the trajectory tracking and obstacle avoidance controller based on control barrier function is applied for moving and static obstacles. The method. Non-innovative, lacking in description and refinement of innovation. The research conclusion lacks credibility.
2. In Section 3.2 ·, there are too few obstacles between the starting point and the goal point during simulation verification, which cannot significantly prove the effectiveness and advancement of the method proposed in this paper.
3. The simulation comparison results in Figure 11 do not show that the method mentioned in this paper is advanced compared with the method in reference 6.
4.The headings in Figures 1 to 3 are the same as: The original planned trajectory will be blocked by obstacles in the next moment. Circles filled with points represent dynamic obstacles, while polygons filled with lines represent the static environment. a) A smooth trajectory given by the motion planner through the clear space safely. b) Two dynamic obstacles.
5. The function name used in Equations 12 and 13 is different from that in the text, the former is h, the latter is B.
Extensive editing of English language required.
Author Response

(The authors gave the same response as above.)

Reviewer 3 Report
The paper titled, “EF-TTOA: Development of a UAV Planner and Avoidance Control Framework for Static and Moving Obstacles” seems to be interesting. Below are Reviewer suggestions and queries. Reviewer insists Authors to address these before considering for publishing.
What is the sensor used to detect obstacles both moving and static? What is the sensor that helps to extract environmental features?
How paths are generate by satisfying the vehicle constraints like turn rate constraint, acceleration constraint, velocity constraint?
Comparison with much more exiting methods with different environment further improve the quality of paper
Reviewer appreciate the simulation with ROS-Gazebo-PX4 SITL environment
Writing requires improvement. For example, in “Abstract”:
As (With) the increasing applications of unmanned aerial vehicles (UAVs) in surveying, mapping, rescue, etc., the security of autonomous flight in complex environments becomes a crucial issue. Many environments include moving objects, such as people and vehicles, which may cause the UAV to crash (in the airspace of UAV how people cause it to crash?? Generally the moving obstacles would be birds and other flying vehicles). However, most of the existing methods cannot deal with a real-time changing environment by using online re-planning. Therefore, the main purpose of this work is to solve the moving obstacles in a common environment by a new planning and control framework for a quadrotor UAV that can quickly extract environmental information and autonomous work safely (this sentence lack clarity kindly rewrite). Firstly, the point clouds acquired from the depth camera is (are) divided into dynamic and static points, and then the velocity of point cloud clusters is estimated. The static point cloud is used as the input of (for ) the local mapping. Path planning is simplified by identifying key points of (better word may be among) static points. Secondly, the design of a trajectory tracking and obstacle avoidance controller based on the control barrier function guarantees security for moving and static obstacles. The path finding module can stably search for the shortest path, and the controller can deal with moving obstacles with high-frequency feedback control. Therefore, the UAV can deal with both long-term planning and immediate emergencies. The framework proposed in this work enables a UAV (to operate ) in a wider field, with better security and more real-time performance.
Reviewer has a suggestions
The title can be written as “EF-TTOA: Development of a UAV Path Planner and Obstacle Avoidance Control Framework for Static and Moving Obstacles”. If authors this suggestion is acceptable they can include it in the paper.
Author Response

(The authors gave the same response as above.)

Reviewer 4 Report
In this paper entitled “EF-TTOA: Development of a UAV Planner and Avoidance Control Framework for Static and Moving Obstacles”, the authors proposed two algorithms to steer a quadrotor UAV, that can quickly extract environmental information, through obstacles in a common environment by a control framework. In my opinion, this is an interesting effort toward developing a robust control system for a UAV. However, I have a few questions and suggestions for the authors.
11. What is the difference between equations 2 & 4? A modification of Equation 2 with the information in Equation 3 is expected to be shown in Equation 4.
22. In Figure 6, the authors mentioned that the red arrows are gradient directions formed by the obstacles. The statement is not clear. Is it the gradient of the trajectory or the surface of the obstacles?

Author Response

(The authors gave the same response as above.)

Round 2
Reviewer 1 Report
Dear Authors,
My concerns on the previous version of the manuscript have been addressed sufficiently. Now, the paper paper is acceptable for possible publication.